# Vitreous Levels of Vascular Endothelial Growth Factor and Platelet-Derived Growth Factor in Patients with Proliferative Diabetic Retinopathy: A Clinical Correlation

**DOI:** 10.3390/biom13111630

**Published:** 2023-11-08

**Authors:** Rami Al-Dwairi, Tamam El-Elimat, Abdelwahab Aleshawi, Ahmed H. Al Sharie, Balqis M. Abu Mousa, Seren Al Beiruti, Ahmad Alkazaleh, Hasan Mohidat

**Affiliations:** 1Division of Ophthalmology, Department of Special Surgery, Faculty of Medicine, Jordan University of Science and Technology, Irbid 22110, Jordan; 2Department of Medicinal Chemistry and Pharmacognosy, Faculty of Pharmacy, Jordan University of Science and Technology, Irbid 22110, Jordan; 3Department of Pathology and Microbiology, Faculty of Medicine, Jordan University of Science and Technology, Irbid 22110, Jordan

**Keywords:** diabetic retinopathy, intravitreal concentrations, vascular endothelial growth factor, VEGF, platelet-derived growth factor, PDGF, visual outcomes

## Abstract

**Background**: The global epidemic status of diabetic retinopathy (DR) and its burden presents an ongoing challenge to health-care systems. It is of great interest to investigate potential prognostic biomarkers of DR. Such markers could aid in detecting early stages of DR, predicting DR progression and its response to therapeutics. Herein, we investigate the prognostic value of intravitreal concentrations of vascular endothelial growth factor (VEGF) and platelet-derived growth factor (PDGF) in a DR cohort. **Materials and methods**: Vitreous sample acquisition was conducted at King Abdullah University Hospital (KAUH) between December 2020 and June 2022. Samples were obtained from any patient scheduled to undergo a pars plana vitrectomy (PPV) for any indication. Included patients were categorized into a DR group or a corresponding non-diabetic (ND) control group. Demographics, clinicopathological variables, standardized laboratory tests results, and optical coherence tomography (OCT) data were obtained for each included individual. Intravitreal concentrations of VEGF and PDGF were assessed using commercial enzyme-linked immunosorbent assay (ELISA). **Results**: A total of 80 eyes from 80 patients (DR group: *n* = 42 and ND control group: *n* = 38) were included in the analysis. The vitreous VEGF levels were significantly higher in the DR group compared to the ND control group (DR group 5744.06 ± 761.5 pg/mL versus ND control group 817.94 ± 403.1 pg/mL, *p* = 0.0001). In addition, the vitreous PDGF levels were also significantly higher in the DR group than those in the ND control group (DR group 4031.51 ± 410.2 pg/mL versus ND control group 2691.46 ± 821.0 pg/mL, *p* = 0.001). Bassline differences between test groups and clinical factors impacting VEGF and PDGF concentrations were investigated as well. Multiple regression analysis indicated PDGF as the sole independent risk factor affecting best-corrected visual acuity (BCVA) at the last follow-up visit: the higher the PDGF vitreous levels, the worst the BCVA. **Conclusions**: Vitreous concentrations of VEGF and PDGF are correlated with DR severity and may exhibit a possible prognostic potential value in DR. Further clinical and experimental data are warranted to confirm the observed findings and to help incorporate them into daily practice.

## 1. Introduction

Diabetes mellitus (DM) is a chronic disease with many long-term complications, including diabetic retinopathy (DR), the world’s leading cause of blindness among adults aged 20–74 years [1]. Although the pathogenesis of DR remains largely unknown, increasing evidence implicates genetics as well as environmental factors [2]. Diabetic patients have a 24-fold increased risk of blindness compared to non-diabetics [3]. DR does not develop in all patients with DM, but chronicity of DM and poor control of blood sugar levels play important roles in the incidence and progression of DR. DR stages are classified according to their severity, from non-proliferative DR (NPDR) to proliferative DR (PDR). Diabetic macular oedema can exist in any stage of the disease and can be focal or diffuse [4,5]. Clinical biomarkers such as glycated hemoglobin A1c (HbA1c), vascular endothelial growth factor (VGEF), platelet-derived growth factor (PDGF), and ocular imaging are used in the diagnosis and prediction of DR as well as monitoring treatment progress [6].

VEGF is a cytokine glycoprotein commonly found as a dimeric molecule. There are several types of cells that can produce and secrete VEGF, including capillary endothelial cells, retinal pigmented epithelium (RPE), Müller cells, astrocytes, and ganglion cells [7,8,9,10,11]. VEGF exerts its physiological effects by mediating vascular permeability, angiogenesis, endothelial cell growth, cell migration, and apoptosis [12]. Seven members are apparently found in the mammalian VEGF family, namely VEGF-A (the prototype of the VEGF family), VEGF-B, VEGF-C, VEGF-D, VEGF-E, VEGF-F, and PlGF (placental growth factor). There are other isoforms of the VEGF family found only in humans, such as VEGF 121, VEGF 165, VEGF 189, and VEGF 206 [13,14]. Increased levels of vitreous and serum VEGF are associated with DR [15]. Therefore, intravitreal injections of anti-VEGF are currently recommended for patients with DR [16]. RPE is a monolayer of pigmented cells located between the retina and choroid of the eye [17]. VEGF is expressed by RPE throughout the ocular development and into adulthood. In adults, RPE appears as the only source of VEGF in the posterior part of the eye. In vitro studies demonstrated that differentiated RPE secretes VEGF in a polarized fashion toward the basolateral side and that VEGFR2 is preferentially expressed by the choriocapillaris on the side facing the RPE [18].

PDGF, which is structurally related to VEGF, is secreted by platelets [19,20]. It is one of the most ubiquitous growth factors that stimulates cellular proliferation and directs cellular movement. Two different PDGF forms exist, PDGF A and PDGF B, which in turn give rise to three PDGF isoforms, namely, PDGF-AA, -BB or -AB [21]. PDGF isoforms have been involved in the stimulation of collagen synthesis and the pathogenesis of proliferative retinal disorders [22]. Patients with PDR have previously been reported to have increased levels of PDGF in their vitreous [20,23].

This study examined VEGF and PDGF-AA vitreous levels in DM-related RD patients and correlated their concentrations with visual and surgical outcomes. This study was undertaken in a community where the DM prevalence is high and the DM control is generally poor.

## 2. Materials and Methods

### 2.1. Patient Recruitment and Sample Acquisition

This study was conducted as per the declaration of Helsinki and approved by the institutional review board (IRB) at King Abdullah University Hospital (KAUH) (IRB number: 58/137/2021). Patient recruitment and surgical sample acquisition were conducted at KAUH, a tertiary referral center for all surgical vitreoretinal cases in Northern of Jordan. The study was conducted between December 2020 and June 2022. Patients who were scheduled to undergo a pars plana vitrectomy (PPV) for any indication were invited to give a vitreous sample during the operation. Demographic data (age and gender), medical history, and comorbidities were collected for each patient. Detailed ocular history for each patient was gathered for number, type, and duration of the last intravitreal anti-VEGF injection, number of panretinal photocoagulation laser sessions, use of anti-glaucoma medications, surgical history, and preoperative details for indicative diagnoses for the PPV. In addition, operative details including description and grading of the retinal findings, associated surgical procedures, and the type of tamponade was obtained as well. Furthermore, serum laboratory investigations (HbA1c and homocysteine) and optical coherence tomography (OCT) were ordered for all the participants at the time of the surgery.

The study included all patients who were scheduled to undergo PPV. Exclusion criteria were a previously vitrectomized patient, recent vitreous hemorrhage (VH) within two months, patients with retinal venous occlusion, where the procedure of obtaining vitreous sample may result in serious complications, prior chemotherapy administration, and when there is a previous cataract surgery complicated by capsular rupture where the fluid could dilute the vitreous. The participants were divided into two groups. The first group, the case group (diabetic retinopathy (DR)), included any patient scheduled for PPV due to advanced PDR complications, which included cases of persistent VH and or fibrovascular membrane (FVM) with tractional retinal detachment (TRD). DR was graded according to the modified Airlie house classifications [24,25]. All cases of advanced PDR were operated either as a case of persistent VH and/or advanced FVM/TRD. The second group, the control group, (non-DR control (N-DR)), included any patient scheduled for PPV due to other indications. These indications comprised rhegmatogenous retinal detachment (RRD), vitreomacular interface diseases (macular hole (MH)), vitreomacular traction (VMT), non-diabetic epiretinal membranes (ERM)), endophthalmitis, and complications following cataract surgery (drop of crystalline lens materials or drop of intraocular lens implants). The control group could include diabetic patients; however, their fundus does not exhibit DR.

Medical history of DM patients (including duration of diabetes, treatment of diabetes, and the level of last HbA1c), hypertension, and chronic kidney diseases were investigated. Intravitreal anti-VEGF injections were used in most patients in the first group preoperatively. They were analyzed and correlated with the level of the biomarkers according to the time and the number of injections given in the first “DR” group. Either aflibercept or ranibizumab were utilized in this study. In addition, the effect of panretinal photocoagulation laser sessions was studied. The possible effect of the associated cataract surgery in some patients during PPV was also studied. Moreover, the type of retinal tamponade after the PPV was collected, which included silicone oil, gas (SF6), and air.

The visual outcome was measured by the best-corrected visual acuity (BCVA) assessed by Snellen decimal projectors. Visual acuity was converted to logMAR visual acuity. For patients with visual acuity of counting fingers, hand motion, light perception, or “no light perception”, they were converted according to the study of Schulze-Bonsel K. et al. [26]. Anterior and posterior segment examination was performed through slit-lamp biomicroscopy with the required non-contact hand-held lenses.

### 2.2. Operative Details and Sample Handling

The PPV operations were performed by a single consultant vitreoretinal surgeon who followed the same procedural guidelines. The same 23-gauge vitrectomy system was utilized for all operations (Combined Wide-Field Elite Pack, Bausch, and Lomb, New York, NY, USA). Under either general or local anesthesia and sterile conditions, three trocars were inserted 4 mm posterior to the limbus. The infratemporal trocar was connected to the infusion cannula; however, the infusion pump was turned off to obtain an undiluted vitreous sample. The second trocar was connected to the light and the last one to the vitreous cutter. The vitreous sample was obtained through the vitreous cutter at a cut rate of 5000 cuts per minute (cpm) from the mid-vitreous while the infusion pump was kept off. This step was done with careful monitoring of the retina. A vitreous sample of 1 to 2 mL was obtained for all participants in a 5 mL syringe that was attached to the vitreous cutter. The required vitreous samples were considered surgical byproducts in which the surgical procedure performed would not be altered or modified, preserving the safety of the patient with optimum surgical efficacy. No additive techniques were implemented for the vitreous sample extraction or collection during vitrectomy.

After that, the infusion pump was turned on and subsequent retinal procedures according to the indications and operative findings were carried out. Additional procedures such as cataract surgery were performed at the beginning as indicated. However, if there was any break in the capsular bag of the lens, the case was excluded as the chance of dilution was presented. For the first “case group”, the VH was cleared initially followed by FVM dissection, segmentation, and delamination. Then, the posterior vitreous surface was detached, and intravitreal injection of triamcinolone was injected. The vitreous body and blood clots in the peripheral vitreous skirt were removed under scleral depression as far as the vitreous base. Intraoperative bleeding was controlled by increasing the irrigation bottle height or endo-diathermy. Endo-laser was applied to complete PRP up to the ora serrata. At the end of the PPV, the decision on the type of tamponade was based on intraoperative findings to prevent the re-detachment of the retina in the postoperative period.

The obtained vitreous samples were transferred to sterile 2 mL Eppendorf tubes. After that, the samples were kept immediately at −80 °C until analyses. The samples were labeled by the names of the participants and the hospital identification numbers and were given assigned numbers.

### 2.3. Sample Processing and Biomarker Quantification

After thawing, the undiluted vitreous samples were centrifuged at 20,000× *g* for 15 min at 4 °C, and the supernatant was then collected. Concentrations of VEGF and PDGF-AA were determined by the enzyme-linked immunosorbent assay (ELISA) in duplicates according to the manufacturer protocol (ABCAM, Cambridge, CB2 0AX, UK). Quality control samples and random analysis of previously quantified samples was performed. The VEGF ELISA was able to detect two of the four VEGF isoforms; VEGF 121 and VEGF 165, which are secreted immature in comparison to longer cell-associated isoforms. The ELISA was carried out using 100 µL aliquots of vitreous, diluted accordingly to comply with the detection range of the relevant assay. The standard solution (100 µL for VEGF, 150 µL for PDGF-AA) was added to the wells of a 96-well plate coated with a monoclonal antibody. After incubation, the plate was washed, and an enzyme-labeled antibody was added followed by the substrate. The reaction was halted by adding the stop solution after the development of color. The optical density of each well was determined by measuring the absorbance at 450 nm and 570 nm using an absorption spectrophotometer (Bio-Tek Instruments, Winooski, VT, USA). A standard curve was generated out of measurements made with the standard solution (12.5–800 pg/mL for VEGF, and 40–30,000 pg/mL for PDGF-AA) and was used to determine the concentration of VEGF or PDGF-AA in each sample. The level of each factor in the vitreous was within the detection range of the relevant assay, with the minimum detectable concentration being 2.7 pg/mL for VEGF (the intra-assay coefficient of variation (CV) was 5.4%, and the inter-assay CV was 5.5%), 4 pg/mL for PDGF-AA (intra-assay CV of 6.3% and inter-assay CV of 6.9%). Protein content was determined in aliquots of vitreous specimens with the standard DC assay (BioRad, Hercules, CA, USA).

### 2.4. Statistical Analysis

Raw data were entered into a spreadsheet and analyzed using the IBM statistical package for the social sciences (SPSS) v.26 (Armonk, New York, NY, USA). In brief, data were expressed as frequency (percentage) or mean ± standard error of the mean (SEM) and statistically analyzed as previously described [27,28]. Logistic regression analysis was implemented to evaluate the predictive potential of VEGF and PDGF vitreous concentrations of visual outcomes; *p*-values ≤ 0.05 were considered statistically significant.

## 3. Results

### 3.1. Study Population

The study included 80 eyes from 80 patients, 45 (56.3%) of whom were males. The mean age of patients was 52.5 years. In 42 patients (52.5%), the right eye was involved. About 65% of patients had both DM and hypertension. Among those with DM, about 60% were treated with insulin-based regimens, and the mean duration of DM was 14.3 years. Mean serum levels of HbA1c and homocysteine were 8.2% and 18.5 mcmol/L, respectively (Table 1).

The DR group (case group) consisted of 42 (52.5%) patients, while the N-DR control group comprised 38 (47.5%). Based on the detailed preoperative diagnoses that necessitated the PPV, all of the patients in the DR group had advanced PDR, including FVM/TRD in 28 cases and persistent VH in 14. In addition, 23 (28.7%) patients had RRD, and 9 (11.3%) patients had vitreomacular interface diseases (ERM/MH/VMT). Moreover, 3 (3.8%) patients had endophthalmitis, while the remaining 3 (3.8%) patients had either dropped intraocular lenses (IOLs) or crystalline lenses as a sequela of complicated cataract surgery (Table 1).

About 38 (90.5%) of the 42 patients in the DR group (case group) received intravitreal anti-VEGF injections preoperatively. The average number of injections was 2.54 and the average number of days from the last injection to the operation was 26.93. Ranibizumab and aflibercept were administered in 26 and 12 patients, respectively. Phacoemulsification with IOL implantation for cataracts was performed on 39 (49.3%) patients. Silicone oil was used as a tamponade in 38 (47.5%) patients, while gas and air were used in 22 (27.5%) and 20 (25%) patients, respectively. Among the total patients, 25 (32.5%) started using AG after the primary operation (Table 1).

### 3.2. Patient Characteristics: DR (Case) vs. N-DR (Control) Groups

The differences in patient characteristics between the DR (case) and N-DR (control) groups are summarized in Table 1. There was a significant difference in age between the DR and N-DR control groups (DR group 55.3 ± 1.4 years versus N-DR group 49.3 ± 2.7 years, *p* = 0.046). In addition, patients in the DR (case) group had significantly more comorbidities, such as DM (DR group 42 cases versus N-DR group 10 cases, *p* = 0.001) and hypertension (DR group 36 cases versus N-DR group 16 cases, *p* = 0.001). In addition, patients in the case group had significantly higher percentage of HbA1c (DR group 9.2% ± 0.3 versus N-DR group 6.1% ± 0.5, *p* = 0.001) (Table 1).

The intravitreal anti-VEGF injections were primarily administered to patients in the DR (case) group. However, four of the ND control group patients who were operated for ERM received intravitreal anti-VEGF injections. It was noticed the DR group had a significantly higher number of cataract surgeries (DR group 30 cases versus N-DR group 11 cases, *p* = 0.0001).

### 3.3. VEGF and PDGF-AA Concentrations in Vitreous Samples

The vitreous VEGF levels were significantly higher in the DR case group compared to the N-DR control group (DR group 5744.06 ± 761.5 pg/mL versus N-DR control group 817.94 ± 403.1 pg/mL, *p* = 0.0001). VEGF/protein levels were also significantly higher in the DR case group than in the N-DR control group (DR group 4133.21 ± 488.2 pg/mg versus N-DR control group 388.95 ± 52.0 pg/mg, *p* = 0.0001) (Table 2, Figure 1A).

The vitreous PDGF-AA levels were also significantly higher in the DR group than those in the N-DR control group (DR group 4031.51 ± 410.2 pg/mL versus N-DR control group 2691.46 ± 821.0 pg/mL, *p* = 0.001). PDGF-AA/protein levels were also significantly higher in the DR group than in the N-DR control group (DR group 2873.08 ± 293.5 pg/mg versus N-DR control group 2007.87 ± 252.3 pg/mg, *p* = 0.03) (Table 2, Figure 1E).

### 3.4. Factors Affecting the Vitreous Levels of VEGF and PDGF-AA

There are a number of factors affecting the levels of VEGF (Table 3) and PDGF-AA (Table 4) in the vitreous of the DR case and N-DR control groups. The vitreous VEGF levels of patients with DM (either in the DR case group or N-DR control group) were found to be higher (4711.95 ± 681.5 pg/mL, *p* = 0.00001). Moreover, higher HbA1c was associated with higher vitreous levels of VEGF (for each one unit increase in HbA1c percentage, the level of VEGF is increased by 771.01 ± 29.8 pg/mL, *p* = 0.013). In addition, as the number of intravitreal anti-VEGF injections increased significantly (for each injection, the levels of VEGF increased by 330 ng/mL). Also, patients with advanced PDR and endophthalmitis have the highest levels of VEGF (5744.06 ± 760.1 and 6181.67 ± 90.7 pg/mL, respectively, *p* = 0.0001), which could be attributed to ischemic changes. Moreover, higher levels of VEGF were found in patients who underwent cataract surgery at the time of PPV (4558.67 ± 456.9 pg/mL, *p* = 0.028). Furthermore, the levels of VEGF were lower when the silicone oil was used as a tamponade. On the other hand, the VEGF levels were not affected by gender, age, laterality, type of intravitreal anti-VEGF, duration of last intravitreal anti-VEGF, or the level of homocysteine. In multiple regression analysis, it was found that being an advanced PDR case is the most important independent factor affecting VEGF levels (Table 3).

Vitreous PDGF levels (Table 4) were affected significantly by the DM treatment regimen. Insulin-based regimen had higher PDGF-AA levels (4281.68 ± 470.3 pg/mL, *p* = 0.02) than oral hypoglycemic regimen. In addition, higher HbA1c levels were associated with higher PDGF-AA levels (for each one unit increase in HbA1c percentage, the level of PDGF-AA is increased by 6.06 ± 0.5 pg/mL, *p* = 0.001). PDGF-AA vitreous levels were also found to be higher in patients with advanced PDR and endophthalmitis as preoperative indications (4031.52 ± 530.8 and 15,301.89 ± 830.4 pg/mL, respectively, *p* = 0.0001). Moreover, patients who underwent cataract surgery had higher vitreous levels of PDGF-AA (4298.03 ± 546.6 pg/mL, *p* = 0.049). As with VEGF, multiple regression analysis indicated that being an advanced PDR case is the most important independent factor affecting PDGF-AA vitreous levels (Table 4).

### 3.5. Visual Outcomes

Patients were followed for visual acuity preoperatively and postoperatively at 1 month, 3 months, and at a last follow-up visit. The correlation between BCVA and the vitreous levels of VEGF (Figure 1A–D) and PDGF-AA (Figure 1E–H) is shown in Figure 1. A significant association was found between PDGF-AA and BCVA across all durations (Figure 1E–H). In general, the higher the level of PDFG-AA the worst the visual acuity. In addition, patients with endophthalmitis and advanced PDR had the worst BCVA, while patients with vitreomacular surface disease (MH/ERM/VMT) had the best visual outcomes. Furthermore, patients who received silicone oil as a tamponade had the worst visual outcomes. Surprisingly, multiple regression analysis indicated PDGF-AA as the sole independent risk factor affecting BCVA at the last follow-up visit: the higher the PDGF-AA vitreous levels, the worse the BCVA.

## 4. Discussion

This study aimed to examine factors affecting the vitreous levels of VEGF and PDGF-AA. This is the first study that assess the VEGF and PDGF in Jordanian and Middle Eastern populations in whom the DM prevalence is high and the control of DM is poor. In addition, this study is characterized by a relatively large sample. It was found that patients in the DR case group (advanced PDR) had higher vitreous levels of both VEGF and PDGF. In addition, VEGF levels were associated with increased levels of serum HbA1c and an increased number of previous intravitreal anti-VEGF injections. Moreover, patients with advanced PDR (TRD/FVM or persistent VH) and endophthalmitis exhibited higher vitreous levels of VEGF and PDGF-AA. This study is the first study to assess VEGF and PDGF-AA levels in patients with endophthalmitis. Furthermore, higher levels of PDGF-AA were associated with worse BCVA, advanced PDR and endophthalmitis, and silicone oil use as a tamponade.

In the process of PDR maturation and progression, VEGF is shown to be a crucial enhancing factor for pathologic retinal neovascularization and FMV maturation [29,30]. Elevated levels of VEGF can promote pathologic transformations of retinal vasculature. However, anti-VEGF treatment inhibits intraocular neovascularization and improves visual function [31]. Vitreous VEGF levels before surgery have been implicated as a risk factor for predicting the outcome or complications of PPV surgery, including early postoperative VH [32,33,34]. PPV is an accepted and effective surgical treatment for advanced PDR. However, pathologic neovascularization and proliferative changes may continue to progress after surgery [35]. Many investigators revealed that the levels of VEGF were significantly reduced in the vitreous of patients with advanced PDR after successful PPV [36]. On the other hand, a high VEGF level may still be maintained in the vitreous cavity after PPV. This suggests that PPV cannot stop VEGF secretion in the vitreous cavity [37]. The results of our study confirm that VEGF levels were much higher in the advanced PDR group as compared to the control group. This is attributed mainly to DR’s ischemic changes. The results of the current study were consistent with those of Wang et al., Praidou et al., Baharivand et al., and Chernykh et al. [20,29,38,39]. Moreover, as the PDR progress and the ischemic areas get worsening, the patients need more intravitreal anti-VEGF. This explains why the increase number of intravitreal anti-VEGF was associated with higher VEGF levels.

Praidou et al. found that PDGF-AA, -AB, and -BB were significantly increased in the vitreous of patients with advanced PDR compared to controls, indicating that PDGF is involved in PDR pathology [20]. PDGF-BB was shown to be increased in diabetic patients compared to non-diabetic patients [32,40]. Praidou et al. also found that levels of PDGF-AA and PDGF-BB in the vitreous correlated significantly only with PDR severity, suggesting that these particular isoforms may contribute to PDR progression [20]. The results of this study provide further evidence that PDGF isoforms are directly associated with PDR pathology. Freyberger et al. reported a significant increase in PDGF-AB concentrations in patients with PDR, with even increased levels in rubeosis iridis eyes [21]. Muhiddin et al. concluded that vitreous PDGF-AB levels were increased in PDR eyes compared with controls. In addition, they provided evidence supporting how anti-PDGF could work on neovascularization processes in PDR [41]. A phase IIb study by Jaffe et al. demonstrated that the combination of anti-VEGF and anti-PDGF was superior to anti-VEGF alone in the treatment of neovascular AMD [42]. Pennock et al. reported in 2014 that VEGF acts via PDGF receptor-α to maintain the sustainability of cells that express both VEGF and PDGF receptors [43]. In the current study, PDGF-AA was significantly increased in eyes with advanced PDR, and it was correlated with higher levels of HbA1c and insulin-based treatment for DM.

RRD induces ischemic changes in the neurosensory retina, which trigger the expression of many growth factors and cytokines in the vitreous and subretinal fluid, such as VEGF, PDGF, hepatocyte growth factor, epidermal growth factor, and fibroblast growth factors [44]. According to Ichsan et al., VEGF and PDGF-AA levels are indicative that RRD surgery should be performed as soon as possible prior to retinal cell death and membrane proliferative formation [45]. Similarly, Rasier et al. reported that patients with RRD had significantly higher levels of VEGF than non-RRD patients [46]. Moreover, Su et al. and Yalcinbayir et al. assessed VEGF levels in the subretinal fluid in patients with RRD and found a significant increase compared to controls [47,48]. However, VEGF and PDGF levels were higher in patients with longer RRD durations. In the current study, VEGF and PDGF-AA levels were significantly lower in the RRD group than in advanced PDR. Most RRD patients were urgently operated within a short period of time.

Another important aspect of the current study of key importance is the relationship between levels of PDGF-AA and visual outcomes. According to our study, BCVA became worse as PDGF levels increased. Mori et al. reported that experimental mice with retinal detachment had higher levels of PDGF-AA [49]. As a mitogen and chemoattractant for RPE cells, retinal glial cells, and pericidal cells, PDGF expression in the eye is increased following RD [49,50]. It has been suggested by Campochiaro et al. that PDGF-AA production by RPE cells increases following RRD. PDGF-AA binds specifically to the PDGFα receptor, which is present in the retinal glial cells and RPE. Consequently, the PDGF-AA level in the eye will increase to a level appropriate for the formation of proliferative membranes [51]. Consequently, it may be postulated that the increased levels of PDGF-AA are associated with more advanced disease. Consequently, the visual outcome could be affected.

It is well known that glycemic control is strongly associated with the incidence and progression of DR. HbA1c is a widely used assay to diagnose DM. It represents the average blood sugar level over a period of two to three months. It is available worldwide and does not require patients to fast. These characteristics make HbA1c one of the most reliable and validated biomarkers used to predict and diagnose diabetic patients with DR [52]. Homocysteine is a non-proteinogenic alpha-amino acid formed by methionine demethylation. It is found mainly in meat and dairy products. Homocysteine may serve as a biomarker for assessing microvascular risk in diabetes and DR [53]. A meta-analysis of 31 studies including 6394 participants found that blood homocysteine concentrations were higher in the DR group compared to the control group, particularly in patients with PDR [54]. In the current study, HbA1c was proportionally correlated with VEGF and PDGF levels. However, serum homocysteine was not correlated with VEGF or PDGF levels.

Silicone oil facilitates retinal reattachment by providing extended intraocular tamponade. In PPV for advanced PDR, anatomical failure and blindness despite the success of PPV in managing the severe complications of diabetic retinopathy may still occur. Recurrence of retinal detachment secondary to fibrovascular proliferation, progression of neovascularization with neovascular glaucoma, and hypotony with subsequent phthisis bulbi, are some of the many reported postoperative complications of diabetic vitrectomy. Accordingly, silicone oil was adopted to be used in the severe forms of advanced PDR [55]. In this study, the use of silicone oil was associated with higher levels of VEGF due to more advanced PDR stages for which silicone oil was used.

Endophthalmitis is defined as an inflammation of the inner coats of the eye, resulting from intraocular colonization of infectious agents with exudation within intraocular fluids (vitreous and aqueous). The most critical factor in the causation of endophthalmitis is the breakdown of the ocular blood barrier and intraocular colonization by pathogens (bacteria/fungi). In exogenous endophthalmitis, the inciting injury or surgery causes disruption of globe integrity, which allows invasion of the pathogens. This is followed by retinal ischemia and necrosis with or without retinal vasculitis [56]. This study was the first to assess VEGF and PDGF-AA levels in vitrectomized due to postoperative endophthalmitis that revealed significantly high levels of VEGF and PDGA-AA, which indicates a marked ischemia in those patients.

## 5. Conclusions

The current study focused on identifying the possible prognostic value of VEGF and PDGF in DR patients. It was found that both of the intravitreal concentrations of VEGF and PDGF are elevated in DR compared to N-DR control group. In addition, both markers (especially PDGF-AA) are correlated with poor visual outcomes with variable predictability power. Moreover, HbA1c control is correlated with the levels of both biomarkers. More studies and trials are needed to justify these results and make a basis for therapeutic regimens.

## Figures and Tables

**Figure 1 biomolecules-13-01630-f001:**
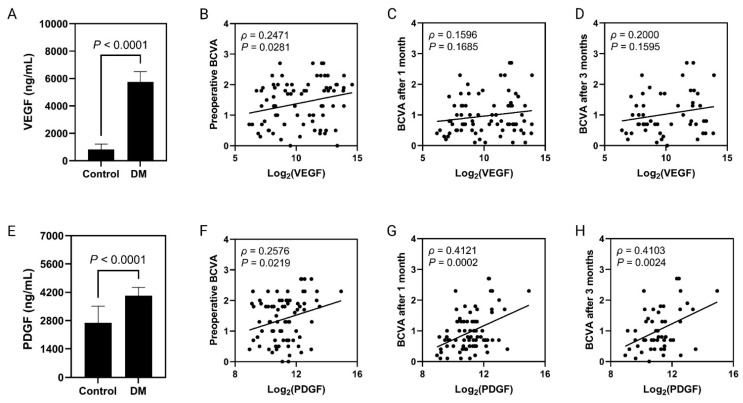
VEGF levels across the DR and ND control group l (**A**). Correlation analysis between intravitreal VEGF concentrations and preoperative (**B**) and postoperative (1 month (**C**) and 3 months (**D**)) BCVA. PDGF levels across the DR and ND control groups (**E**). Correlation analysis between intravitreal PDGF concentrations and preoperative (**F**) and postoperative (1 month (**G**) and 3 months (**H**)) BCVA.

**Table 1 biomolecules-13-01630-t001:** Demographical details of the study population (*n* = 80) and comparison between diabetic retinopathy and control groups.

Variables	Number (Percentage) or Mean ± SEM			
	Overall (*n* = 80)	DR Case Group (*n* = 42)	N-DR Control Group (*n* = 38)	*p*-Value
**Age (years)**	52.5 ± 1.5	55.3 ± 1.4	49.3 ± 2.7	0.046
**Gender**				
Male	45 (56.3)	20 (47.6)	25 (65.8)	NS
Female	35 (43.8)	22(52.4)	13 (34.2)	
**Eye laterality**				
Right eye (OD)	42 (52.5)	20 (47.6)	22 (57.9)	NS
Left eye (OS)	38 (47.5)	22 (52.4)	16 (42.1)	
**Comorbidities**				
Diabetes mellitus	52 (65.0)	42 (100)	10 (26.3)	0.001
Hypertension	52 (65.0)	36 (58.7)	16 (42.1)	0.001
Chronic kidney disease	5 (6.3)	4 (6.6)	12 (10.9)	NS
**Treatment of DM**				
Oral hypoglycemic agents	20 (39.2)	15 (36.6)	5 (50)	NS
Oral hypoglycemic agents and/or insulin	31 (60.8)	26 (36.4)	5 (50)	
**Duration of diabetes mellitus (years)**	14.3 ± 1.1	15.1 ± 1.1	10.6 ± 2.8	NS
**Type of intravitreal injection**				
Aflibercept	12 (31.6)	10 (29.4)	2 (50)	NS
Ranibizumab	26 (28.4)	24 (70.6)	2 (50)	
**Number of intravitreal injections**	2.5 ± 0.4	4.2 ± 0.7	0.7 ± 0.4	0.00001
**Time elapsed since last intravitreal injection (days)**	26.9 ± 9.8	18.2 ± 8.2	91.8 ± 50.4	0.013
**Preoperative diagnosis**				
Advanced PDR (FVM and VH)	42 (52.5)	42 (100)	0 (0)	0.0000
Rhegmatogenous retinal detachment	23 (28.7)	0 (0)	23 (60.5)	
Vitreomacular interface diseases (ERM/MH/VMT)	9 (11.3)	0 (0)	9 (23.7)	
Endophthalmitis	3 (3.8)	0 (0)	3 (7.9)	
Dropped IOL or crystalline lens	3 (3.8)	0 (0)	3 (7.9)	
**Associated cataract surgery** **with the primary PPV**				
Yes	39 (49.3)	30 (71.4)	11 (28.9)	0.0001
No	41 (50.7)	12 (28.6)	27 (71.1)	
**Type of vitreous tamponade**				
Silicone oil	38 (47.5)	15 (35.7)	23 (60.5)	0.035
Gas	22 (27.5)	12 (28.6)	10 (26.3)	
Air	20 (25)	15 (35.7)	5 (13.2)	
**Postoperative AG use**	25 (32.5)	11 (28.2)	14 (36.8)	NS
**Follow-up period (months)**	17.2 ± 1.3	16.0 ± 2.1	17.8 ± 1.8	NS
**Visual outcomes (Log MAR)**				
Preoperative BCVA	1.40 ± 0.1	1.5 ± 0.1	1.3 ± 0.1	NS
BCVA after 1 month	0.97 ± 0.1	1.00 ± 0.1	0.9 ± 0.1	NS
BCVA after 3 months	1.03 ± 0.1	1.2 ± 0.1	0.9 ± 0.1	NS
BCVA at the last follow-up	0.95 ± 0.1	1.0 ± 0.1	0.9 ± 0.1	NS
**HbA1c (%)**	8.2 ± 0.3	9.23 ± 0.3	6.06 ± 0.5	0.001
**Homocysteine (mcmol/L)**	18.5 ± 1.9	18.0 ± 2.0	19.22 ± 3.5	NS

**Table 2 biomolecules-13-01630-t002:** Measured biomarkers (VEGF and PDGF-AA) in vitreous samples in both the control and the diabetic retinal detachment groups.

Measured Markers	Number (Percentage) or Mean ± SEM		
	DR Case Group (*n* = 42)	N-DR Control Group (*n* = 38)	*p*-Value
VEGF (pg/mL)	5744.06 ± 761.5	817.94 ± 403.1	0.0001
VEGF/Protein (pg/mg)	4133.21 ± 488.2	388.95 ± 52.0	0.0001
PDGF-AA (pg/mL)	4031.51 ± 410.2	2691.46 ± 821.0	0.001
PDGF-AA/protein (pg/mg)	2873.08 ± 293.5	2007.87 ± 252.3	0.03
Protein content (mg/mL)	1.56 ± 0.1	2.00 ± 0.4	NS

**Table 3 biomolecules-13-01630-t003:** Factors affecting the level of VEGF.

Variables	Mean ± SEM * or B Regression Coefficient ± SEM **	
	VEGF (pg/mL)	*p*-Value
**Age (years) ****	−18.1 ± 3.8	NS
**Gender ***		
Male	2832.30 ± 674.9	NS
Female	4139.40 ± 807.7	
**Eye laterality ***		
Right eye (OD)	3438.86 ± 740.1	NS
Left eye (OS)	3365.80 ± 648.2	
**Comorbidities**		
**Diabetes mellitus ***		
Yes	4711.95 ± 681.5	0.00001
No	975.39 ± 545.8	
**Hypertension ***		
Yes	3908.65 ± 647.4	NS
No	2467 ± 456. 6	
**Treatment of DM ***		
Oral hypoglycemic agents	3769.47 ± 828.2	NS
Oral hypoglycemic agents and/or insulin	5202.27 ± 1001.3	
**Duration of diabetes mellitus (years) ****	165.30 ± 12.1	NS
**Type of intravitreal injection ***		
Aflibercept	4682.29 ± 1030.5	NS
Ranibizumab	5899.54 ± 1092.6	
**Number of intravitreal injections ****	330 ± 13.0	0.013
**Time elapsed since last intravitreal injection (days) ****	−16.30 ± 1.3	NS
**Preoperative diagnosis ***		
Advanced PDR (FVM and VH)	5744.06 ± 760.1 ↑	0.0001
Rhegmatogenous retinal detachment	418.16 ± 57.2	
Vitreomacular interface diseases (ERM/MH/VMT)	259.43 ± 46.1	
Endophthalmitis	6181.67 ± 90.7 ↑↑	
Dropped IOL or crystalline lens	194.67 ± 22.1	
**Associated cataract surgery** **with the primary PPV ***		
Yes	4558.67 ± 456.9	0.028
No	2261.05 ± 789.0	
**Type of vitreous tamponade ***		
Silicone oil	2025.74 ± 520.3	0.036
Gas	4373.68 ± 1045.1	
Air	4956.66 ± 1391.6	
**Postoperative AG use ***	2405.15 ± 567.4	NS
**HbA1c (%) ****	771.01 ± 29.8	0.013
**Homocysteine ****	−10.36 ± 4.2	NS

* The test used was ANOVA and data are presented as mean and standard error. ** The test used was linear logistic regression and data are presented as B coefficients and standard error.

**Table 4 biomolecules-13-01630-t004:** Factors affecting the level of PDGF-AA.

Variables	Mean ± SEM * or B Regression Coefficient ± SEM **	
	PDGF-AA (pg/mL)	*p*-Value
**Age (years) ****	−57.23 ± 3.2	NS
**Gender ***		
Male	2879.33 ± 314.8	NS
Female	4057.99 ± 939.3	
**Eye laterality ***		
Right eye (OD)	3016.33 ± 567.2	NS
Left eye (OS)	3813.52 ± 346.8	
**Comorbidities**		
**Diabetes mellitus ***		
Yes	3583.70 ± 360.7	NS
No	3044.5 ±990.7	
**Hypertension ***		
Yes	3629.90 ± 613.1	NS
No	2958.7 ± 679.1	
**Treatment of DM ***		
Oral hypoglycemic agents	2556.37 ± 516.2	0.02
Oral hypoglycemic agents and/or insulin	4281.68 ± 470.3	
**Duration of diabetes mellitus (years) ****	93.73 ± 6.2	NS
**Type of intravitreal injection ***		
Aflibercept	4066.29 ± 723.1	NS
Ranibizumab	3540.43 ± 546.1	
**Number of intravitreal injections ****	−12.78 ± 11.6	NS
**Time elapsed since last intravitreal injection (days) ****	−6.74 ± 0.6	NS
**Preoperative diagnosis ***		
Advanced PDR (FVM and VH)	4031.52 ± 530.8 ↑	0.0001
Rhegmatogenous retinal detachment	1641.44 ± 101.1	
Vitreomacular interface diseases (ERM/MH/VMT)	1830.65 ± 83.1	
Endophthalmitis	15,301.89 ± 830.4 ↑↑	
Dropped IOL or crystalline lens	713.61 ± 99.6	
**Associated cataract surgery** **with the primary PPV ***		
Yes	4298.03 ± 546.6	0.049
No	2517.79 ± 789.1	
**Type of vitreous tamponade ***		
Silicone oil	3766.45 ± 643.9	NS
Gas	2571.87 ± 567.1	
Air	3594.66 ± 989.5	
**HbA1c (%) ****	6.06 ± 0.5	0.001
**Homocysteine ****	19.22 ± 3.5	NS

* The test used was ANOVA and data are presented as mean and standard error. ** The test used was linear logistic regression and data are presented as B coefficients and standard error.

## Data Availability

Data are available upon request from the corresponding author.

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
