# Peer review of "Vitreous Levels of Vascular Endothelial Growth Factor and Platelet-Derived Growth Factor in Patients with Proliferative Diabetic Retinopathy: A Clinical Correlation"

_biomolecules, 2023, doi:10.3390/biom13111630_

Round 1

Reviewer 1 Report

Comments and Suggestions for Authors

The manuscript Al-Dwairi et al. titled “The Prognostic Value of Intravitreal Concentrations of Vascular Endothelial Growth Factor (VEGF) and Platelet-derived Growth Factor (PDGF) in Proliferative Diabetic Retinopathy” is an interesting article. However, the following issue needs to be addressed before the manuscript is recommended for publication.

1. As the Authors stated, the VEGF is secreted in various retinal components, however, references 7-11 in the Introduction section are insufficient. Especially, the authors should introduce the importance of the retinal pigment epithelium for maintaining ocular homeostasis including secretion of VEGF, since the breakdown of RPE cause various ocular diseases including AMD and DR. The reviewer recommends introducing RPE related to VEGF secretion with proper citation. The recommended references are the following,

[1] Kim, et al. "Development of 3D printed Bruch’s membrane-mimetic substance for the maturation of retinal pigment epithelial cells." International journal of molecular sciences 22.3 (2021): 1095.

[2] SAINT-GENIEZ, Magali, et al. An essential role for RPE-derived soluble VEGF in the maintenance of the choriocapillaris. Proceedings of the National Academy of Sciences, 2009, 106.44: 18751-18756.

2. The authors summarized data obtained from 42 DR patients and 38 ND control patients and compared their characteristics between DR and ND control. However, as stated in Table 1, there is an age difference between these two groups, with a meaningful P-value. This can be interpreted that age could be a factor for affecting the concentration of VEGF and PDGF as shown in Table 2.

3. Is the patient group different between Table 1 and 3? Especially, in age section, the age is stated as 18.1 and 57.23 in VEGF and PDGF respectively. Did the samples are collected from different patients between VEGF and PDGF samples? In addition, the age of the patients for VEGF sample collecting is too different compared with other data. The author must provide detailed information for collecting data for Tables 1, 2, and 3.

4. The concentration of VEGF in patients treated with anti-VEGF is very high considering 5744.06 pg/ml in DR group of Table 2 and 4139.4 in females of Table 3. The authors must deeply discuss this phenomenon in the discussion section.

5. There are 8 graphs in Figure 1, but only 1A and 1E were cited in section 3.3.

Author Response

Editor-in-Chief

Biomolecules

Re: The Prognostic Value of Intravitreal Concentrations of Vascular Endothelial Growth Factor (VEGF) and Platelet-derived Growth Factor (PDGF) in Proliferative Diabetic Retinopathy. Manuscript ID: biomolecules-2630064

Dear Editor and Reviewers,

My self and the co-authors are pleased to resubmit the manuscript entitled ‘The Prognostic Value of Intravitreal Concentrations of Vascular Endothelial Growth Factor (VEGF) and Platelet-derived Growth Factor (PDGF) in Proliferative Diabetic Retinopathy” to be considered for publication in Biomolecules. The revised manuscript takes into consideration both the editorial and reviewers’ comments. Kindly find below a point-by-point response to those comments, along with an uploaded copy marked with MS track changes indicating changes to the manuscript.

Response to reviewer 1

Many thanks for your valuable comments and remarks, the authors will address each comment separately, and alterations will be amended in the revised manuscript accordingly:

  • "As the Authors stated, the VEGF is secreted in various retinal components, however, references 7-11 in the Introduction section are insufficient. Especially, the authors should introduce the importance of the retinal pigment epithelium for maintaining ocular homeostasis including secretion of VEGF, since the breakdown of RPE cause various ocular diseases including AMD and DR. The reviewer recommends introducing RPE related to VEGF secretion with proper citation. The recommended references are the following," Thank you very much for your comprehensive comment. An elaborative introduction for the role of RPE in VEGF metabolism was added for the Introduction with the proper citations.
  • “The authors summarized data obtained from 42 DR patients and 38 ND control patients and compared their characteristics between DR and ND control. However, as stated in Table 1, there is an age difference between these two groups, with a meaningful P-value. This can be interpreted that age could be a factor for affecting the concentration of VEGF and PDGF as shown in Table 2.” Thank you very much again for this important point. For this type of analysis, age matching is difficult to be obtained as the diabetic patients tend to be older. In addition, in table 3, we performed a linear logistic regression analysis to for the effect of age on VEGF and PDGF concentrations which indicated no relationship.
  • “ Is the patient group different between Table 1 and 3? Especially, in age section, the age is stated as 18.1 and 57.23 in VEGF and PDGF respectively. Did the samples are collected from different patients between VEGF and PDGF samples? In addition, the age of the patients for VEGF sample collecting is too different compared with other data. The author must provide detailed information for collecting data for Tables 1, 2, and 3.” Thank you very much for this point. The data provided in table 3 is not the age, it is the B coefficient for the age, which indicates the increase (if +) or the reduction (if –) of VEGF or PDGF concentrations for one unit of age. We have clarified this in the revised manuscript. Thank you again.
  • “The concentration of VEGF in patients treated with anti-VEGF is very high considering 5744.06 pg/ml in DR group of Table 2 and 4139.4 in females of Table 3. The authors must deeply discuss this phenomenon in the discussion section.” Thank you very much for your meaningful comment. First, not all patients in the DR had received anti-VEGF. However, it is proven from table 3 that for each injection, the level of VEGF is increased by 330 pg/ml. This is explained by the fact that the more advanced cases of PDR with more progressive ischemia generally need more anti-VEGF. Regarding the second point, the females in general had more VEGF and PDGF concentration as indication in table 3. This can be explained from table 1 as the DR group had more females and the non-DR group had more males. All these findings were discussed in the revised manuscript.
  • “There are 8 graphs in Figure 1, but only 1A and 1E were cited in section 3.3.” Thank you. The correction was done in the revised manuscript.

Reviewer 2 Report

Comments and Suggestions for Authors

In this study the authors measured vitreal levels of VEGF and PDGF proteins collected during vitrectomy and show that both are elevated in the eyes of diabetics with severe proliferative retinopathy. These are then related to different additional factors that may influence the levels of VEGF and PDGF.  Regressions are also done to determine how VEGF and PDGF levels are correlated with acuity both before and after vitrectomy.  I believe that the presentation of the results needs work, and that the authors need to clarify why they did the study and what new findings came from it.

1. It is not clear what is new here. The authors cite several other studies that have found similar results, that is, elevated VEGF and PDGF, and they do not claim anything new or different from previous work.  VEGF in particular is well known to be a major player in proliferative retinopathy. The introduction is very general and does not specify a hypothesis or what the authors hoped to contribute with these studies, and the discussion similarly does not discuss anything that is claimed to be new in these results.  If the finding that only high PDGF is related to worse post-vitrectomy acuity is new, this should be indicated.  There are other things that might be new, for instance a correlation between vitreal VEGF and HbA1C, but this is not clear.

2. The authors need to explain why these data are of prognostic value.  The abstract says that further data are needed “to help incorporate the information into clinical practice.”  Assuming the best case, how would data like this inform clinical practice?  Are the authors proposing that PDGF could be measured with a vitreal sample before deciding to do surgery, and that if it is high, vitrectomy should not be done?  Or are the authors arguing that a sample taken during vitrectomy could be helpful in followup treatment?  How would it help with followup treatment. PDGF might help predict visual outcome, but would this influence the patient treatment post-vitrectomy?  Biomarkers, such as HbA1c or imaging, should be measurable with minimally invasive methods, so I do not think that these proteins fit that role.  Even if there would be some potential value of PDGF measurement, the data suggest that it would have very poor specificity and sensitivity, because the scatter in PDGF levels at the same pre- and post-acuity value is very large. However, while it might not be helpful clinically, it might be still might advance research to know that there is a relationship between PDGF and acuity.

3. The authors attempted to take prior treatments into account as well as such factors as hypertension, but table 3 mixes regression slopes with levels of VEGF and PDGF in a way that is confusing. I think the regression slopes should be put in a different table than the levels in pg/ml, and for regressions we should be given R2. There are other questions also.  Are the male, female, right eye, left eye, hypertension,  values in this table the overall averages, including DM and control?  Overall averages do not seem useful.  It might be useful to have male vs. female (or OS/OD) differences for just the diabetic population. It looks like some of the values in the table are for diabetics, others are for controls, and some are mixed. When there are tests of these values, it is not always clear what the comparison group is. Sometimes it seems like it must be comparison of two adjacent values in the table, but sometimes the comparison group does not seem to be listed. For instance, what was the 3908 pg/ml for hypertension compared to in order to conclude that it was NS?

4. Line 236: the VEGF level in the DR group is given as 5744 pg/ml, which agrees with table 1, but on line 254 what appears to be the same parameter is given as 4711.95.

5. Line 255: This says that higher HbA1c was associated with higher VEGF, but the level in parentheses, 771, is not a high level of VEGF, so what is that value?  If there is higher VEGF with higher HbA1c, that would be reflected in a regression analysis, and what we would need to know is the strength of that correlation. Since the point about HbA1c also appears in the discussion, it would be worth showing a plot of Hb1Ac vs. VEGF.  There is a similar issue on line 269 with PDGF,

6. The “overall” column in table 2 does not seem useful.

7. Line 256: It seems odd that with more anti-VEGF injections the level of VEGF would be higher. Do the authors think that injections actually increase VEGF level or is there some other explanation for this correlation?  Do the VEGF protein measurements differentiate between free VEGF and VEGF that is inactivated by a drug, or are both of these incorporated in the VEGF level?

8. Line 267: “Insulin-based regimen had higher PDGF-AA levels...” Compared to what? Is this a comparison of diabetics with controls or a comparison of insulin-treated diabetics with those not taking insulin? If the latter, it seems puzzling that PDGF would be higher in insulin-treated diabetics, unless those not treated had some other factor that could explain the difference.

9. Figure 1.  I do not understand why base 2 logarithms are used for VEGF.  And, from the scatter plots in B,C, and D, it looks like the average log2(VEGF) is about 10.  The average VEGF is about 5700 (from A), and the log of this base 2 is about 12.5, but it does not look like the average value of VEGF in B,C, and D would be 12.5. Please clarify.

Comments on the Quality of English Language

Only minor problems with English

Author Response

Editor-in-Chief

Biomolecules

Re: The Prognostic Value of Intravitreal Concentrations of Vascular Endothelial Growth Factor (VEGF) and Platelet-derived Growth Factor (PDGF) in Proliferative Diabetic Retinopathy. Manuscript ID: biomolecules-2630064

Dear Editor and Reviewers,

My self and the co-authors are pleased to resubmit the manuscript entitled ‘The Prognostic Value of Intravitreal Concentrations of Vascular Endothelial Growth Factor (VEGF) and Platelet-derived Growth Factor (PDGF) in Proliferative Diabetic Retinopathy” to be considered for publication in Biomolecules. The revised manuscript takes into consideration both the editorial and reviewers’ comments. Kindly find below a point-by-point response to those comments, along with an uploaded copy marked with MS track changes indicating changes to the manuscript.

Response to reviewer 2

Many thanks for your valuable comments and remarks, the authors will address each comment separately, and alterations will be amended in the revised manuscript accordingly:

  • “It is not clear what is new here. The authors cite several other studies that have found similar results, that is, elevated VEGF and PDGF, and they do not claim anything new or different from previous work.  VEGF in particular is well known to be a major player in proliferative retinopathy. The introduction is very general and does not specify a hypothesis or what the authors hoped to contribute with these studies, and the discussion similarly does not discuss anything that is claimed to be new in these results.  If the finding that only high PDGF is related to worse post-vitrectomy acuity is new, this should be indicated.  There are other things that might be new, for instance a correlation between vitreal VEGF and HbA1C, but this is not clear.” Thank you very much for your efforts. This study is similar to other studies as you indicated. However, this is the first study to be done on Jordan where the prevalence of DM is very high, and the control of DM is generally poor. In addition, the sample size is relatively larger than other studies. Moreover, the variability of the studied diagnoses in the control group (non-DR) is another important point. In addition, it the first study to assess the level of the biomarkers in patients with endophthalmitis. Regarding the findings, basically it is supposed to have the similar pattern of other studies. However, the new findings will be highlighted more in the discussion. Collectively, we think that this study may add to the literature. An elaboration in the Introduction and Discussion was done in the revised manuscript to highlight the new aspect for this study. Thank you very much for your meaningful comment.
  • “The authors need to explain why these data are of prognostic value. The abstract says that further data are needed “to help incorporate the information into clinical practice.”  Assuming the best case, how would data like this inform clinical practice?  Are the authors proposing that PDGF could be measured with a vitreal sample before deciding to do surgery, and that if it is high, vitrectomy should not be done?  Or are the authors arguing that a sample taken during vitrectomy could be helpful in followup treatment?  How would it help with followup treatment. PDGF might help predict visual outcome, but would this influence the patient treatment post-vitrectomy?  Biomarkers, such as HbA1c or imaging, should be measurable with minimally invasive methods, so I do not think that these proteins fit that role.  Even if there would be some potential value of PDGF measurement, the data suggest that it would have very poor specificity and sensitivity, because the scatter in PDGF levels at the same pre- and post-acuity value is very large. However, while it might not be helpful clinically, it might be still might advance research to know that there is a relationship between PDGF and acuity.” Thank you so much. We agree with that. At the near future, it might be helpful clinically, however, further research is needed to justify these results first. Then, different outcomes should be correlated with the PDGF level before judging any clinical acuity correlation. Thank you so much, we have corrected this in the revised manuscript.
  • “ The authors attempted to take prior treatments into account as well as such factors as hypertension, but table 3 mixes regression slopes with levels of VEGF and PDGF in a way that is confusing. I think the regression slopes should be put in a different table than the levels in pg/ml, and for regressions we should be given R2. There are other questions also. Are the male, female, right eye, left eye, hypertension,  values in this table the overall averages, including DM and control?  Overall averages do not seem useful.  It might be useful to have male vs. female (or OS/OD) differences for just the diabetic population. It looks like some of the values in the table are for diabetics, others are for controls, and some are mixed. When there are tests of these values, it is not always clear what the comparison group is. Sometimes it seems like it must be comparison of two adjacent values in the table, but sometimes the comparison group does not seem to be listed. For instance, what was the 3908 pg/ml for hypertension compared to in order to conclude that it was NS?” Thank you so much for your meaningful comment and important notice. Table 3 summarizes the factors that may affect the VEGF and PDGF levels. Two tests were conducted, the first one is ANOVA test which was used when the factor is categorical (such as sex, diagnosis), and the values for this analysis are the differences not the overall average. These values were presented by mean and standard error. On the other hand, linear regression analysis test was used when the factor is numerical in nature (such as age, HBA1C level), and the data was presented and studied by the B coefficient with standard error which reflect the change of VEGF or PDGF levels by the change of one unit of the factor. All comparisons were made for the whole sample (not diabetic or control alone) except factors that are found only in diabetic (such as duration of DM, number of injections, and HB1c). Regarding factors such as HTN, and DM, the comparison group are the group that does not have the disease.
    We agree that the data in table 3 are confusing, accordingly, we will split the table into VEGF alone and PDGF alone. In addition, we will clarify each test used in the analysis either ANOVA test or linear regression analysis test. Moreover, we will clarify the factors such HTN, and DM and insert the MF groups. Thank you so much.
  • “ Line 236: the VEGF level in the DR group is given as 5744 pg/ml, which agrees with table 1, but on line 254 what appears to be the same parameter is given as 4711.95.” Thank you so much. The analysis in line 254 is regarding the presence of DM not DR. It is important to notice that 10 participants in the control group have well-controlled DM without the presence of DR as indicated in table 1. The control group is defined as patients who underwent PPV for advanced PDR. This point will be confirmed more in the revised manuscript.
  • “Line 255: This says that higher HbA1c was associated with higher VEGF, but the level in parentheses, 771, is not a high level of VEGF, so what is that value? If there is higher VEGF with higher HbA1c, that would be reflected in a regression analysis, and what we would need to know is the strength of that correlation. Since the point about HbA1c also appears in the discussion, it would be worth showing a plot of Hb1Ac vs. VEGF.  There is a similar issue on line 269 with PDGF” Thank you very much. This value is the B coefficient of the regression analysis, which means that for an increase of one percent of HbA1c, the VEGF level is elevated by 771 pg/ml. We will clarify this in the revised manuscript.
  • “ The “overall” column in table 2 does not seem useful.” Thank you very much. We have omitted this column.
  • “ Line 256: It seems odd that with more anti-VEGF injections the level of VEGF would be higher. Do the authors think that injections actually increase VEGF level or is there some other explanation for this correlation? Do the VEGF protein measurements differentiate between free VEGF and VEGF that is inactivated by a drug, or are both of these incorporated in the VEGF level?” Thank you very much. This correlation is true and it indicates that patients who needed more anti-VEGF injections are actually have more advanced disease and more advanced ischemia
  • “ Line 267: “Insulin-based regimen had higher PDGF-AA levels...” Compared to what? Is this a comparison of diabetics with controls or a comparison of insulin-treated diabetics with those not taking insulin? If the latter, it seems puzzling that PDGF would be higher in insulin-treated diabetics, unless those not treated had some other factor that could explain the difference.” Thank you much for your respectful comment. It compared to patients who did not take insulin. As with the previous comment, patients who treated with insulin are more likely to have more advanced disease.
  • “Figure 1. I do not understand why base 2 logarithms are used for VEGF.  And, from the scatter plots in B,C, and D, it looks like the average log2(VEGF) is about 10.  The average VEGF is about 5700 (from A), and the log of this base 2 is about 12.5, but it does not look like the average value of VEGF in B,C, and D would be 12.5. Please clarify.” Thank you so much for your significant notice. The binary logarithm is used in this case of numbers in thousands. This figure plots the relation between the BCVA and Log2 for VEGF. It indicates that the average is actually near 12.5 in relation to the BCVA. Thanks again.

Many thanks again for your time, looking forward to publish with your esteemed journal. Please, if there are additional modifications, we are ready for any revision.

Again, we thank the editor and reviewers for their constructive comments. We hope that we have sufficiently addressed their concerns.

Sincerely,

Authors

Reviewer 3 Report

Comments and Suggestions for Authors

Good research work. The conclusions are interesting for a possible therapeutic idea.

Author Response

Good research work. The conclusions are interesting for a possible therapeutic idea." Thank you very much.

Round 2

Reviewer 1 Report

Comments and Suggestions for Authors

The authors revised the manuscript according to their revision, however, some of the revised parts must be addressed. 

What is the AA1 ~ AA4?

Is it revised or added new references?

If that, the authors must state the information of newly added references.

Comments on the Quality of English Language

It seems that the manuscript requires an English editing.

Author Response

Again and again, many thanks for your efforts in improving our manuscript

We are glad for that

"What is the AA1 ~ AA4?

Is it revised or added new references?

If that, the authors must state the information of newly added references."

These phrases (AA1- AA4) appear in the PDF version not the word version. It is actually the place for our comments regarding the place for the new references and the new information were highlighted by the word tracker. So, it is not an error that present in the revised manuscript. 

English language editing was done.

Many thanks 

Reviewer 2 Report

Comments and Suggestions for Authors

The paper is considerably improved and some important aspects that were puzzling have been clarified.  The main comment that the authors did not address concerns the idea of prognostic value of measurements of intraocular VEGF and PDGF proteins. In the best case, if all the results are confirmed in other studies, how could measurements of these proteins be of prognostic value?  Would the authors base a decision to do a vitrectomy on the pre-vitrectomy values of VEGF or PDGF?  Would this guide treatment or recommendations to the patient after vitrectomy?  If the authors cannot even make a suggestion about how these measurements would be of prognostic value, then they should delete the first 4 words of the title of the paper, and not refer to prognostic value elsewhere.  I think the results are likely to be informative in a research context, but not have much prognostic value. 

Also, I do not really understand why 4 of the 10 diabetics in the control group were in some cases treated with anti-VEGF agents if they did not have DR.  I think they must have had at least background retinopathy, but perhaps they received these treatments in an attempt to treat a comorbidity?

Comments on the Quality of English Language

Most of the insertions in the text in answer to reviewer comments need English correction.  There are also a few cases where the paper contains what seems to be a leftover comment from an MS Word version (for instance AA1).

Author Response

Thank you very much dear reviewer. Your efforts in improving the manuscript are well appreciated

"

The paper is considerably improved and some important aspects that were puzzling have been clarified.  The main comment that the authors did not address concerns the idea of prognostic value of measurements of intraocular VEGF and PDGF proteins. In the best case, if all the results are confirmed in other studies, how could measurements of these proteins be of prognostic value?  Would the authors base a decision to do a vitrectomy on the pre-vitrectomy values of VEGF or PDGF?  Would this guide treatment or recommendations to the patient after vitrectomy?  If the authors cannot even make a suggestion about how these measurements would be of prognostic value, then they should delete the first 4 words of the title of the paper, and not refer to prognostic value elsewhere.  I think the results are likely to be informative in a research context, but not have much prognostic value. " Thank you, we agree with that regarding this point. However, the results of this study can affect the practice by suggesting new promising treatment such as Anti-PDGF agent, or can predict the outcome of the possible PPV, or can even encourage the patient to optimize their risk factors. Thank you very much and we have modified the title. 

"Also, I do not really understand why 4 of the 10 diabetics in the control group were in some cases treated with anti-VEGF agents if they did not have DR.  I think they must have had at least background retinopathy, but perhaps they received these treatments in an attempt to treat a comorbidity?" Thank you very much. These 4 patients have an Idiopathic ERM in which anti-VEGF injection may be a line of treatment before PPV.

"

Most of the insertions in the text in answer to reviewer comments need English correction.  There are also a few cases where the paper contains what seems to be a leftover comment from an MS Word version (for instance AA1)."

Thank you very much. An English language editing was done. 

Many thanks for you